# Allograft Inflammatory Factor-1 in Metazoans: Focus on Invertebrates

**DOI:** 10.3390/biology9110355

**Published:** 2020-10-24

**Authors:** Jacopo Vizioli, Tiziano Verri, Patrizia Pagliara

**Affiliations:** 1Inserm, Univ.Lille, Inserm, U1192—Protéomique Réponse Inflammatoire Spectrométrie de Masse—PRISM, F-59000 Lille, France; jacopo.vizioli@univ-lille.fr; 2Dipartimento di Scienze e Tecnologie Biologiche e Ambientali, Università del Salento, Via Provinciale Lecce-Monteroni, 73100 Lecce, Italy; tiziano.verri@unisalento.it

**Keywords:** AIF-1, Iba1, immunity, macrophages, invertebrates, inflammation, bacterial challenge

## Abstract

**Simple Summary:**

During their life, all living organisms defend themselves from pathogens using complex strategies. Vertebrates and invertebrates share mechanisms and molecules that guarantee their overall bodily integrity. Allograft inflammatory factor-1 (AIF-1) is a protein extensively studied in vertebrates, and especially in mammals. This factor, generally involved in inflammation events occurring upon pathogenic infection or tissue injury, is linked to several important human diseases. This review collects data on the presence and role of AIF-1 in invertebrates, which are still poorly investigated organisms. Multiple alignment and phylogenetic analysis reveal that AIF-1 is conserved in vertebrates and invertebrates, suggesting similarity of functions. In some invertebrate species, the expression of AIF-1 increases considerably after a bacterial challenge, indicating that it plays a key role during the immune responses. This review highlights the importance of studying this protein in invertebrates as a way to improve our knowledge of innate immunity mechanisms and to better understand inflammatory regulation events in mammals.

**Abstract:**

Allograft inflammatory factor-1 (AIF-1) is a calcium-binding scaffold/adaptor protein often associated with inflammatory diseases. Originally cloned from active macrophages in humans and rats, this gene has also been identified in other vertebrates and in several invertebrate species. Among metazoans, AIF-1 protein sequences remain relatively highly conserved. Generally, the highest expression levels of *AIF-1* are observed in immunocytes, suggesting that it plays a key role in immunity. In mammals, the expression of *AIF-1* has been reported in different cell types such as activated macrophages, microglial cells, and dendritic cells. Its main immunomodulatory role during the inflammatory response has been highlighted. Among invertebrates, *AIF-1* is involved in innate immunity, being in many cases upregulated in response to biotic and physical challenges. *AIF-1* transcripts result ubiquitously expressed in all examined tissues from invertebrates, suggesting its participation in a variety of biological processes, but its role remains largely unknown. This review aims to present current knowledge on the role and modulation of *AIF-1* and to highlight its function along the evolutionary scale.

## 1. Introduction

Living organisms protect themselves from pathogens or tissue injury through a complex regulatory network of processes, among which inflammation plays an important role [1]. Inflammation represents the first response of the immune system to harmful stimuli and is a crucial event to initiate the healing process. When tissues are injured by bacteria, trauma, toxins, heat, or any other cause, the damaged cells release molecules responsible for leaking fluid from the blood vessels into the tissues. This causes a swelling that helps isolate the foreign substance from further contact with body tissues. The released molecules also attract phagocytes able to “eat” foreign particles or damaged cells. The main purpose is the elimination or inactivation of the intruders.

During the inflammatory process, the coordinated activation of various signaling pathways regulates inflammatory mediator levels in resident tissue cells and inflammatory cells recruited from the blood [2]. In this sense, deregulated inflammatory states can drive many chronic diseases, including cardiovascular and bowel diseases, diabetes, arthritis, and cancer [3].

Allograft inflammatory factor-1 (AIF-1) is a calcium-binding scaffold/adaptor protein often associated with inflammatory diseases. *AIF-1* was originally cloned from active macrophages in rat and human atherosclerotic allogenic heart grafts undergoing chronic transplant rejection [4,5]. AIF-1 belongs to a family of proteins including three other molecules identical to AIF-1, which were reported under the names of MRF-1, Iba1, and Daintain, respectively, and a series of proteins sharing a large, but not total, identity with AIF-1 (i.e., IRT-1, BART-1, and G1) [6]. Subsequently, AIF-1 has been identified and characterized in many other species, so we can consider it as one of the most evolutionarily conserved inflammatory genes. AIF-1 is a 17-kDa protein with a central pair of EF-hand motifs [7]. This feature is common to a large family of Ca^2+^-binding proteins known as EF-hand proteins [8]. Moreover, the AIF-1 protein and gene are also known as ionized calcium-binding adapter molecule 1 (Iba1), a 147-amino-acid Ca^2+^-binding protein widely used as a marker for microglia [9]. AIF-1 is an important regulator of immune response and is involved in a large panel of inflammation-associated pathologies. An overview of inflammatory diseases mediated by AIF-1 was published by Zhao et al. [10]. In this work, the authors report the involvement of this factor in allograft rejections, vasculopathies, autoimmune diseases, central nervous system (CNS) injuries, and cancers, enhancing its role as a regulator of inflammatory mediators such as iNOS, cytokines, and chemokines.

In this review, we will present, after a short mention of AIF-1 in vertebrates, an overview of this factor as identified and characterized in phylogenetically distant invertebrate species, including sponges, cnidarians, mollusks, annelids, and echinoderms.

The more we learn about these distantly related ancestors, the more we realize how much we share with them. To date, it is possible to find several disparate, and often unrelated, studies on the expression and role of *AIF-1* within different invertebrate species. Furthermore, it is worth considering the role and modulation of AIF-1 in the immune response of invertebrate organisms as well as its functional and structural conservation along metazoan evolution. A thorough understanding of ancient immune systems will not only help us to identify chinks in the armor of invertebrate pests but also provide a window to the better comprehension of our own innate immune mechanisms.

## 2. Vertebrata

All multicellular organisms, vertebrates or invertebrates, possess an immune system that is an essential component of the defense strategies to recognize and neutralize parasites, microorganisms, viruses, and more. Vertebrates have two lines of defense, defined as innate, or non-specific, immunity and adaptive, or acquired, immunity. The success of the immune response is guaranteed by a complex interweaving of interactions between different types of molecules, each with specific functions.

In mammals, the expression of *AIF-1* has been reported in different cell types such as activated macrophages, microglial cells, and dendritic cells (DC). The main immunomodulatory role of the protein during the inflammatory response has also been highlighted in these cells. However, this gene plays different roles in the nervous and immune systems [11,12]. Furthermore, *AIF-1* is also expressed in muscle, liver, spleen, and thymus in rats [13] and humans [14], and it is considered a marker of activated human vascular smooth muscle cells and arterial injury [15]. A possible link between skeletal muscle cell proliferation and AIF-1-induced inhibition of satellite cell proliferation has also been revealed [14], expanding the possible fields of action of this protein. In 2017, Elizondo et al. underlined the importance of AIF-1 in antigen presentation by DC [16]. In this study, they reported that *AIF-1* is expressed in CD11c^+^ dendritic cells and that expression silencing restrains induction of antigen-specific CD4^+^ T cell effector responses. Moreover, because *AIF-1* knockdown in murine DC resulted in impaired T cell proliferation, the same authors demonstrated that *AIF-1* expression in DC serves as a potent governor of cognate T cell responses [17]. Furthermore, Miyata et al. [18] showed an upregulation of *AIF-1* transcripts in red seabream (Teleostean) leukocytes upon LPS stimulation and suggested a similar function in Vertebrata. More recently, a novel role of AIF-1 as a Ca^2+^-responsive scaffold protein involved in cell differentiation emerged [19]. In particular, the requirement of *AIF-1* expression in hematopoietic progenitors for differentiation into Mo-DC and cDC1 subsets has been evidenced.

The AIF-1 protein represents a crucial element for macrophages’ survival and pro-inflammatory activity [20,21]. It is involved in inflammation and immune responses associated with autoimmune diseases [22], and also with vasculopathy [23] and CNS injury [24]. In a recent review, Sikora et al. [25] summarized the role of AIF-1 in the pathogenesis of some diseases including endometriosis, breast cancer, atherosclerosis, rheumatoid arthritis, and fibrosis. Indeed, several authors evidenced its importance in rheumatoid arthritis progression [26,27,28]. *AIF-1* is also considered as a new risk factor for the development of atherosclerosis [29], and, when overexpressed, it influences the intensification of atherosclerotic plaque calcification [30]. Recent studies report a tight link between AIF-1 and cancer. It is involved in breast cancer development by interacting with several proteins such as metalloproteinases [31] or transcription factors [32] and activating the downstream pathways, inducing cell proliferation. Interestingly, *AIF-1* is also expressed in the CNS of vertebrates. Mostly known under the name of Iba1, this factor is largely recognized as a specific microglial marker allowing the distinction of these brain-resident immune cells from neurons and other brain glial cells [9]. *AIF-1* is modulated upon different brain injuries and pathologies, indicating a link with CNS inflammatory states [10]. *AIF-1* expression is generally linked to the presence of activated brain microglia/macrophages. Beschorner et al. [33] reported the expression of *AIF-1* in a limited subpopulation of microglial cells and did not observe a significant upregulation of this gene upon traumatic brain injury. In contrast, Schwab et al. [34] observed an important accumulation of AIF-1^+^ cells in microglia/macrophages in association with experimental spinal cord injury. Indeed, the authors indicate this accumulation as essential for the initiation of an effective response to CNS injury and repair events. Interestingly, *AIF-1* expression has been reported in human microglia following cerebral infarction [35]. This suggests that *AIF-1* can also be upregulated by non-inflammatory brain lesions such as hypoxia.

## 3. Invertebrata

Despite the limited amount of published data, it appears that in vertebrates, like in vertebrates, *AIF-1* is mainly involved in the inflammatory response. To date, *AIF-1* genes have been characterized functionally in phylogenetically distant group of species, such as sponges, cnidarians, mollusks, annelids, and echinoderms. Many invertebrate species are of commercial interest, and more and more research groups are directing their investigations to these animals and their defense mechanisms. Aquaculture of numerous species suffers from massive infections and severe mortality. Therefore, it is necessary to understand and solve disease-related problems. Furthermore, due to the relative simplicity of their immune system, invertebrates also represent good models for studying innate immunity and providing insight into the evolution of their defense mechanisms.

Invertebrates are characterized by a lack of acquired immunity; thus, in such organisms the innate immune systems can provide the host with an immediate defense against pathogens in a non-specific manner. During this event, the immune system is able to differentiate self from non-self. The recognition process is carried out by circulating cells named amebocytes, hemocytes, or coelomocytes according to the various animal groups. In many invertebrate species, these cells have a macrophage-like appearance and function. They are characterized by the presence on their surface of pathogen recognition receptors (PRRs) recognizing pathogen-associated molecular patterns (PAMPs). These are well-conserved molecular structures expressed by various pathogens whether they are viruses, bacteria, or other foreign particles. The receptor–ligand binding triggers a complex cascade of cellular reactions with the production of a wide array of effector molecules.

AIF-1 is a ubiquitously expressed and well-conserved molecule involved in innate immunity response from sponges [36,37] to humans [6]. Notably, when compared all together, AIF-1-related proteins of vertebrates and invertebrates easily branch according to the major metazoan groups. In addition, AIF-1-related proteins are also detectable in choanoflagellates, i.e., the closest living relatives of animals (Figure 1A). As summarized in Figure 1B, percent identity values remain invariably high (never lower than 40%) among metazoan groups. For details on the actual percent identity values, see also Appendix A.

In spite of the evolutionary distance of the selected species, multiple alignment of the AIF-1 sequences from vertebrates, invertebrates, and choanoflagellates reveals highly conserved motifs and structural features (Figure 2). In particular, AIF-1 protein sequence lengths span from 142 to 158 amino acids, with major sequence differences mainly discernable at the amino- and carboxy-terminal ends. Multiple sequence alignment shows the presence of 19 fully conserved amino acid residues (asterisks) and 32 positions exhibiting high conservation, thanks to the presence of amino acids with strongly similar properties (colons). The core of the proteins invariably contains two adjacent EF-hand (EFh) calcium-binding motifs, the second being less conserved than the first. Canonically, this type of domain consists of a 12-residue loop flanked on both sides by a 12-residue α-helical domain. Ca^2+^ binding induces a conformational change in the EFh motif, leading to the activation or inactivation of target proteins.

### 3.1. Porifera

Porifera are the phylogenetically oldest still existent metazoans in which *AIF-1* has been identified [36,37]. In these organisms, the protein presents a high sequence similarity with vertebrates. Kruse et al. [36] observed that in the sponges *Suberites domuncula* and *Geodia cydonium* the expression of *AIF-1* mRNA was induced in cytokine-mediated allogeneic responses during wound repair. Interestingly, its expression does not occur in autografts, suggesting a possible function in immunocytes involved in alloimmune rejection [36]. Cloning of *S. domuncula AIF-1* allows demonstration that the distribution of the six exon/intron borders is, with one exception, strictly conserved between sponge, human, and mouse genes. This also suggests a close evolutionary distance between these species [37]. Moreover, in the same sponge it has been documented, both at tissue and in vitro level, that the expression of *AIF-1* and certain Tcf-like transcription factor genes is closely correlated with histoincompatibility reactions [37]. Indeed, *AIF-1* expression is upregulated in transplants, especially in grafts deriving from different donors. According to Kruse et al. [36], these data suggest that in sponges, in addition to an adaptive immunity, an effector system involving a cytokine-mediated activation of immunocytes occurs. Although we do not know which cells produce AIF-1 in sponges, it is possible to hypothesize that the ‘gray cells’, which are equivalent to the immune leukocytes within the vertebrates, are the main AIF-1 producers [37].

### 3.2. Cnidaria

Apart from a gene sequence from *Nematostella vectensis* present in GenBank (Acc. No. XP_001635454), data on AIF-1 in Cnidaria only derive from the sea anemone *Anemonia viridis*, where an AIF-1 homologue was first identified and characterized [38]. The predicted protein shows the common elements of AIF-1 family members, possessing the evolutionarily conserved EF-hand Ca^2+^-binding motifs, the typical post-transcriptional modification sites, and a 3D structure that can be superimposed with human members of this family [38]. Probably, in *A. viridis* AIF-1 serves as a general protective factor under normal physiological conditions. However, after challenges with different stresses (i.e., biotic or physical challenge) a transcriptional activation can be observed, confirming the involvement of AIF-1 in the inflammatory response [38]. In this anthozoan species, AIF-1 transcripts are detected in different tissues including the tentacles, oral disk, body wall, pharynx, and basal disk. Considering the basic diblastic organization of Cnidaria, these results evidence that in *A. viridis* the expression pattern of AIF-1 could be distributed in the cells of both the ectoderm and endoderm layers [38]. However, it is not clear at present whether the presence of this protein is limited to immunocytes, which possibly infiltrate the tissues.

### 3.3. Mollusca

Among invertebrates, due to their commercial interest, mollusks represent the most investigated species. Furthermore, frequently suffering from environmental bacterial infection, investigations in mollusks are prevalently targeted at the identification of genes modulated upon bacterial challenge. The first report of *AIF-1* in mollusks was in *Haliotis diversicolor* hemocytes, where *AIF-1* was one of the 34 genes involved in different cellular pathways upregulated after bacterial challenge [39]. Moreover, in this gastropod a high expression level of *AIF-1* mRNA was found in gills, suggesting that it could also have a significant contribution in the prevention of microbial infection [40].

AIF-1 cDNA has also been cloned from Bivalves including oysters (*Pinctada martensii*, *Crassostrea gigas*, and *Crassostrea ariakensis*) [41,42,43], triangle sail mussel (*Hyriopsis cumingii*) [44], scallops (*Chlamys farreri*) [45], and clams (*Venerupis philippinarum*) [40], and also from gastropods such as abalones (*Haliotis discus discus*) [46]. Upregulation of *AIF-1* has been observed in some species after tissue injury such as tissue implant in *H. cumingii* [44] or shell damage and mantle injury in *P. martensii* [41]. In all these species, AIF-1 molecules were active in the host immune responses against pathogenic challenges. Indeed, in *V. philippinarum*, *P. martensii*, and *H. discus discus* this gene was upregulated in hemocytes after infection by both Gram-positive and Gram-negative bacteria [40,41,46], indicating its possible role in clearing pathogens soon after the infection. In support of this, there are also data showing the increased expression of *AIF-1* after LPS stimulation in *H. cumingii* [44] and *C. ariakensis* [43]. In the latter mollusk, using a recombinant AIF-1 protein, Xu et al. [43] highlighted that AIF-1 acts in the regulation of some immune-related genes such as *LITAF*, *MyD88*, and *TGFβ*. Interestingly, LITAF is an important transcription factor and is believed to regulate the expression of inflammatory-related factors IL-1α, TNFα, and IFN-γ in mammals [47,48].

Studies on the role of AIF-1 in *H. cumingi* evidenced a significant increase in the phagocytosis rate [45]. This is not surprising since mollusk hemocytes, the main component of cellular immune responses in invertebrates, can be considered functionally analogous to vertebrate leukocytes, acting as macrophages and playing a crucial role in the recognition and removal of foreign materials [49]. Recently, the functional role in hemocytes has been better pointed out by confocal imaging, which revealed that AIF-1 regulates phagocytosis via a functional interaction with filamentous actin [50]. In hemocytes, AIF-1 appeared diffused in the cytoplasm and colocalized with F-actin bundles. After a bacterial challenge, a disruption of the AIF-1 and F-actin association and an increase in cell extension occurred. In all the investigated mollusks, the *AIF-1* gene resulted constitutively expressed in various tissues such as mantle, gill, hepatopancreas, muscle, and foot, with the highest level always being recorded in hemocytes [40,46]. Some authors report the constitutive expression of *AIF-1* transcripts in a variety of unstimulated tissues. This is probably due to its involvement in various processes other than inflammation, pathogenic challenges, or tissue injury, which still need to be explored. However, although AIF-1 has been found in several organs, we cannot currently say whether it is expressed in cells other than hemocytes generally present in various tissues.

Gust et al. [51] used the *AIF-1* gene as a marker to evaluate the immune effects of environmentally relevant concentrations of pharmaceutical mixtures on the pond snail *Lymnaea stagnalis*. Results indicate that this factor, together with other immune and inflammatory markers, is modulated in the presence of drug mixtures and municipal effluent water in this gastropod. In particular, the *AIF-1* gene in *L. stagnalis* is downregulated in response to antibiotic and psychiatric drug mixtures. This study demonstrates the interest of *AIF-1* as a potential biomarker for environmental studies on water chemical pollution.

### 3.4. Annelida

AIF-1 was initially characterized in annelids by Drago et al. [52], who reported its presence, under the name of Iba1, in the CNS of the medicinal leech *Hirudo medicinalis*. Iba1 is a largely recognized microglial marker in vertebrates, though it has not been detected in parenchymal brain microglia of zebrafish and birds [53]. According to the authors, this work constitutes the first report of such a factor in the CNS of an invertebrate species [52]. Like its vertebrate counterpart, the leech gene *Iba1*/*AIF-1* is upregulated in nervous cells upon experimental injury or ATP stimulation. The predicted Iba1 protein shows an average identity of about 50% and 55% with AIF-1 proteins described in vertebrates and invertebrates, respectively. Immunohistochemistry analyses demonstrated its presence in activated microglia accumulated at the injury site of connective fibers and in those surrounding neuron cell bodies. Results from Schorn et al. [54] established the constitutive presence of the Iba1/AIF-1 protein in CD68+ and CD45+ macrophage-like cells spread in leech body wall tissues. The number of AIF-1 immunopositive cells strongly increases upon bacterial challenge. In addition, the injection of recombinant AIF-1 in leeches induces massive angiogenesis and, similarly to AIF-1 in vertebrates, promotes macrophage-like cell recruitment at the injured site. Recent works from the same team [55,56] demonstrated that RNASET2, a protein belonging to the T2 ribonuclease family, and AIF-1 are released from the same immunocompetent cells in the leech *H. verbana8*. Both factors would contribute to the recruitment of immune cells (granulocytes and macrophages) upon LPS injection or wound healing, resulting in the activation of an effective response against pathogen infection. AIF-1 has also been identified in *Hirudo* telocytes [57], special resident cells involved in the immune-surveillance system of the leech body wall. Like in vertebrates, these cells also play a role in regulating immune and neuroendocrine functions in leeches. All together, these data indicate the involvement of AIF-1 in immune cell activation and migration as well as in regulating the inflammatory response in leeches.

### 3.5. Echinodermata

Echinoderms represent the most developed invertebrates and the bridge leading to the primitive chordates, cephalochordates, and urochordates, in which many autologous genes and functions from their ancestors can be found.

The evolutionary position of echinoderms among the Deuterostomia, the same evolutionary branch where vertebrates are found, represents one of the reasons that makes these animals attractive as upcoming model systems. Echinoderms, besides being an important source of food and medicine for humans, provide important clues to understand immune functions that are common with vertebrates. Firstly, Elie Metchnikoff, by introducing the comparative approach to immunology [58], postulated the inflammation concept applicable to mammals and other animals with closed circulatory systems. Metchnikoff’s study in this field started from the observation of phagocyte recruitment around foreign material in the larvae of an echinoderm (sea star) [59].

Only many years later were the immune and non-self recognition capabilities established in these organisms [60,61,62,63] when molecular approaches were developed to evidence novel immune effectors.

The first report on the presence of *AIF-1* in echinoderms [64] was based on the EST analysis of genes upregulated in coelomocytes in response to LPS challenge. Indeed, Nair et al. [64] identified one gene, named *Sp1086*, matching allograft inflammatory factor-1 in the purple sea urchin, *Strongylocentrotus purpuratus*.

Within the Echinodermata phylum, the sea urchin represents an excellent model organism for studies on inflammation, including those on the expression and regulation of *AIF-1*. Indeed, the Antarctic sea urchin *Sterechinus neumayeri* [65] and the common sea urchin *Paracentrotus lividus* [66,67] have furnished intriguing information about the association of *AIF-1* with the immune response. AIF-1 has been identified in coelomocytes of *S. neumayeri*, where an increase in expression during the first phase of the immune response to a bacterial challenge was evidenced. Interestingly, with *S. neumayeri* being a sea urchin living in circumpolar waters, the protein primary structure presents some molecular adaptations to cold [65], and this report indicates that *AIF-1* can participate in the inflammatory response in extremely cold environments.

Further data have been derived from investigation in the common sea urchin *P. lividus*, where the molecular identification and functional characterization of *AIF-1* have been recently reported [68]. In this work, the authors found a significant increase in *AIF-1* expression, at both the mRNA and protein level, in coelomocytes after Gram+ bacterial challenge. In addition, immunocytochemical analysis conducted on different coelomocyte populations revealed the presence of the AIF-1 protein in the perinuclear cytoplasmic zone of amoebocytes and inside red sphaerula cell granules. With these cells being involved in the inflammatory reaction, it is possible to support that AIF-1 plays a crucial role in the defense processes within echinoderms. More recently, information on the *AIF-1* gene in *P. lividus* has been enriched by the work of Chiaramonte et al. [67], who reported its modulation following LPS challenge and the bioinformatically characterized protein structure.

In the sea cucumber *Apostichopus japonicus*, the full-length cDNA of *AIF-1* has been cloned [68]. Like the sea urchin, in this organism a significant increase in the expression levels of *AIF-1* transcripts has also been detected in coelomocytes after bacterial challenge and papilla injury. Based on these results, the authors supported the idea that AIF-1 is involved in acute inflammatory response. Furthermore, data reported the constitutive expression of *AIF-1* in all the tested tissues, including body wall, intestine, respiratory tree, tube feet, and longitudinal muscle.

## 4. Conclusions

Genes belonging to the AIF-1 family have been observed in many metazoan phyla. The sequences of the predicted proteins display a high level of conservation, particularly concerning the structure of the two EF-hand Ca^2+^-binding regions typical of this factor. Interestingly, although *AIF-1* genes have been described in phylogenetically distant species, they have not been identified in yeast or plants and are also absent in some Protostomia, such as in *Drosophila melanogaster* or *Caenorabditis elegans*, the most studied animal models within the arthropod and nematode groups, respectively.

From a phylogenetic/evolutionary perspective, we evidence the steadily increasing numbers of allograft inflammatory factor-like (nucleotide and amino acid) sequences from invertebrates now available in the various databases/databanks online. A systematic functional genomics approach is now probably required to analyze, rationalize, and possibly re-classify the gene/gene product(s) assortment detectable in the genomes of the various metazoan species. This will also help in re-considering, re-assessing, and better defining the roles and functions of the various AIF-1 genes and proteins. In addition, better knowledge of the role of AIF-1/Iba1 in inflammatory pathways might lead to the identification of new therapeutic targets for some neuroinflammatory diseases [69].

AIF-1 activity is clearly linked to inflammation and immune response events in all the animal phyla in which this protein has been characterized. The spectrum of action and the involvement of this factor in the immune response of metazoans need to be further investigated. In invertebrates, in particular, *AIF-1* is generally upregulated in hemocytes after bacterial challenge, confirming its involvement in inflammatory/immune processes. Indeed, its presence in a large variety of invertebrates’ organs and tissue could be explained by the infiltration of circulating hemocytes. Another common feature of most cells expressing *AIF-1* is their ability to migrate upon inflammatory signals [69]. This suggests that the AIF-1 protein might play a crucial role in cell mobility events, possibly regulating the calcium metabolism of activated cells. Although the AIF-1 protein is generally present in cells belonging to the monocyte/macrophage lineage, it can also be associated with other very peculiar cell types, such as the elongated spermatids present in the luminal aspect of mouse testes [70]. AIF-1 presence is correlated with spermatid development and differentiation stages, suggesting that this protein could be involved in the reorganization of the actin cytoskeleton during spermatogenesis and cytoplasmic residue elimination, occurring in the final stage of spermiogenesis.

Invertebrates constitute a largely unexplored source of experimental models. Future studies on *AIF-1* in these organisms will bring new knowledge on its biological functions, which are probably not relegated to inflammatory regulation. It could also open up unexpected insights for the comprehension of AIF-1 functions in mammal immune response and the control of human inflammatory diseases.

## Figures and Tables

**Figure 1 biology-09-00355-f001:**
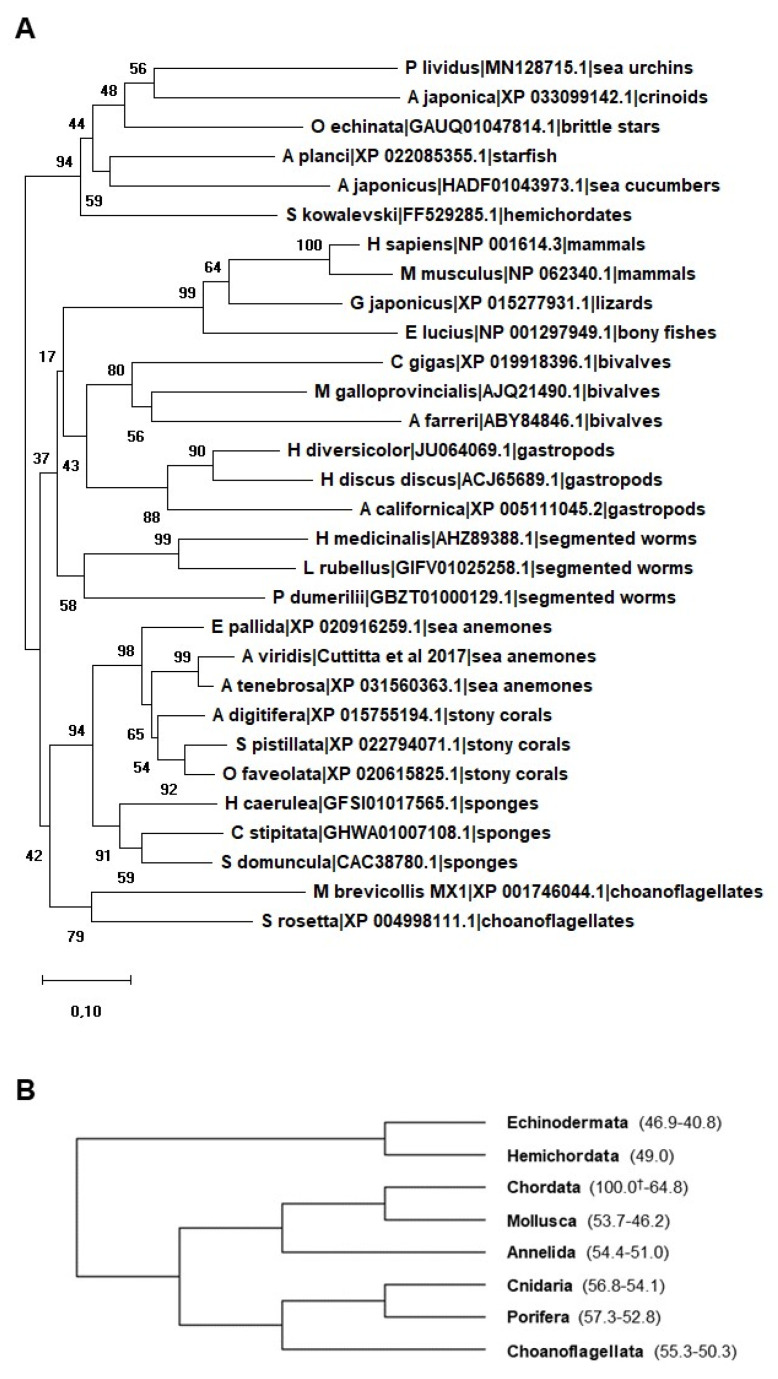
(**A**) Neighbor-joining (NJ) optimal tree based on selected metazoan allograft inflammatory factor-1 (AIF-1)-related proteins. The tree was generated using MEGA X including AIF-1 from different species ranging from mammals to porifera. The amino acid sequences of two choanoflagellates are also included in the tree. All the sequences used were obtained from GenBank at NCBI. The evolutionary history was inferred using the NJ method. The evolutionary distances were computed using the Poisson correction method and are in the units of the number of amino acid substitutions per site. Bootstrap test: 1000 replicates. All ambiguous positions were removed for each sequence pair. (**B**) Condensed tree of (**A**) showing relationships among taxa. Percent identity values (min-max values amongst those observed in each metazoan group) are indicated in parentheses. † indicates *Homo sapiens* AIF-1 reference identity value (100%).

**Figure 2 biology-09-00355-f002:**
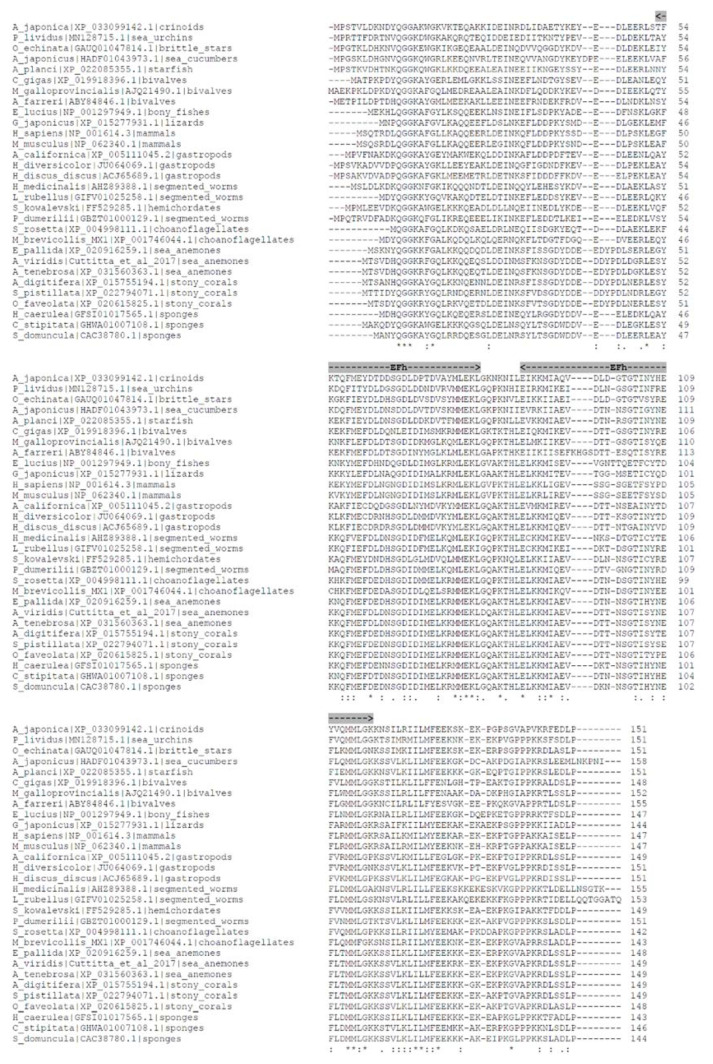
Multiple alignment among the sequences analyzed in Figure 1 as obtained by Clustal Omega (default parameters). EF-hand, calcium-binding motifs (Efh) are highlighted in grey. Asterisks (*) correspond to single, fully conserved residues. Colons (:) indicate conservation of residues sharing strongly similar properties (equivalent to scoring > 0.5 in the Gonnet PAM 250 matrix). Periods (.) indicate conservation of residues with weakly similar properties (equivalent to scoring ≤ 0.5 and > 0 in the Gonnet PAM 250 matrix).

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
