# Peer review of "Allograft Inflammatory Factor-1 in Metazoans: Focus on Invertebrates"

_biology, 2020, doi:10.3390/biology9110355_

Round 1

Reviewer 1 Report

Title: The allograft inflammatory factor-1 in Metazoans

Authors: Jacopo Vizioli , Tiziano Verri , Patrizia Pagliara *

MAJOR POINTS:

The authors aim to provide a concise review on allograft inflammatory factor-1 in the context of evolutionary conservation within metazoans. This is particularly important given the disparate and often non-connected studies of AIF1 within different species -- but that the general consensus being that gene plays a dominant role in some form of immunity. This review follows on the heels of a recently published work by Pawlik A et al (PMID: 31830499) and differentiates itself by comparing/contrasting the different metazoan groups. This reviewer feels that this review article will provide an important interconnective context of the role of AIF1 in general immunity, and serve as a good introductory review for evolutionary biologists and ecologists with a principal interest in studying immunity in respective target species.

Several areas are incorrect throughout the manuscript that must be corrected/clarified. For example, the work introduces AIF1 as a cytokine -- that is wholeheartedly and inherently incorrect. It is an intracellular scaffold/adaptor protein, not a cytokine. For example, there is no report of a receptor for the AIF1 that is used for immune responses. There are several other areas that are also incorrect, including limitations in monocytes/macrophages (which is not true) and how AIF1 is regulated. The manuscript does do well by identifying the several metazoan species to which it is expressed and pointing out published findings -- but some of the presented notions are incorrect or misinterpreted. This, as a review, is a major concern -- as the purpose is to prevent reported data correctly in a compendium format (i.e. this is not an opinion piece). As such, much work needs to go into correcting the verbiage and presentation of work before it can be presented as a valuable review article.  

KEY POINTS:

  1. Line 10-12, Line 34, Line 233: AIF1 is NOT a cytokine. It is a 17kDa calcium binding scaffold/adaptor protein that is intracellular in function; it is not secreted as a soluble factor (no work has corroborated the poorly done studies by the group that made the sole claim, whereas all other 100+ publications have shown it is intracellular in function as a signaling modulator)
  2. Line 13: unclear what the authors mean by “similarity of the identified amino acid sequences” reach high percentage values … Are the authors trying to simply state: “Among metazoans, there are relatively high conservation protein sequences”. Generally, we refer to conserved gene sequences at the DNA level -- as opposed to amino acid. For amino acid changes, that would be important if there were pivotal changes in substitutions that confer differential function.
  3. Line 15 “AIF1 observed in circulating cells” is NOT a correct statement. Circulating cells would also include red blood cells, but they surely don’t express AIF1  -- why not use the correct term “leukocytes” or “immunocytes”.
  4. Line 17, 45 - several works have shown dendritic cells, which are a distinct lineage that monocyte/macrophages, also prominently express the gene. This must be noted.  
  5. Line 18-19 - This is not a correct statement. AIF-1 is upregulated upon *any* inflammatory stimuli, whether it drive TLR-agonist or cytokine related. In fact, AIF-1 was originally described as an IFN gamma-responsive element…  
  6. Line 19-21 - This statement is misleading and thereby incorrect. AIF1 is NOT present in all examined tissue. See the protein tissue atlas: https://www.proteinatlas.org/ENSG00000204472-AIF1/tissue
  7. Line 32-33 - It is misleading to state inflammation is a common pathogenesis state. Authors should say “deregulated inflammatory states can drive many chronic diseases…” or something akin
  8. Line 51 - AIF1 does NOT govern proliferation in cells expressing the gene. The cited work by the authors in the sentence notes that disruption of AIF1 in macrophages acts upon/affects neighboring cell proliferative capacity.
  9. Line 71-74 - It is important to note that microglial are not “macrophage-like”, but are actually derived from common macrophage-precursors established within the neuroectoderm.  Their transcriptomic signature warrants their role. Also, it is expressed in all microglial in any system with a CNS (i.e. all vertebrates have a CNS) - so it is all vertebrates.
    1. Fate mapping analysis reveals that adult microglia derive from primitive macrophages. Ginhoux F, Greter M, Leboeuf M, Nandi S, See P, Gokhan S, Mehler MF, Conway SJ, Ng LG, Stanley ER, Samokhvalov IM, Merad M. Science. 2010 Nov 5;330(6005):841-5. doi: 10.1126/science.1194637. Epub 2010 Oct 21.
  10. Line 76 - Authors begin by defining invertebrates as having a lack of acquired immunity. However, the authors never established in the vertebrate section that another hallmark of the close circulatory system of vertebrates allows adaptives and innate immunity. That should be the opening line within section 2:vertebrates
  11. Line 80 - Unclear what the authors mean by “AIF1-type” proteins. Is that the possible genes with close sequence homology to that of AIF1. If so, that should be clearly explained, as to not be confused with AIF1-like protein (gene id: 83543; AIF1L). Suggest to use the word “related” as opposed to “type”, to better convey gene homology as determining criteria for the argument of conservation.
  12. Line 81-83 - The statement “with respect to vertebrate the role of 81 AIF-1 in invertebrates has been poorly documented, it appears that also in invertebrates AIF-1 is 82 mainly involved in the inflammatory response.” … this suggests that AIF1- has other roles beyond inflammatory control in vertebrates. This has not been documented/shown. AIF1 is involved in inflammatory-immune responses, with the two mutually integrative. The lack of adaptive immunity does not preclude the principal role in modulation of immune responses, which is associated with inflammation, among other downstream effectors.
  13. Line 109 - It is unclear what the authors mean by “cell-mediated recognition system”. Is this adaptive immunity (TCR:pMHC)? If so, should use this terminology, as cell-mediated recognition is obscure and ambiguous.
  14. Line 111 - “it is possible to hypothesize that it is the ‘gray cells’ featuring vertebrate immune cells” -- This reviewer understands what the author is trying to state, but it is not explained well.  The author means to say “it is possible to hypothesize that the gray cells, which are equivalent to innate immune leukocytes within the vertebrate,”... as the principal producers of AIF1
  15. Line 115-118 - Citation required for the statement (i.e. who shows that superimposition on human members…?”
  16. Line 122-125 - The statement is misleading. Although AIF1 is found within the diverse tissues, no clear evidence has been shown which cell types they are in. For example, it can be present in oral disk tissues, but the oral disk is composed of functional and sentinel (immunocytes) cells; similar to noting on line 141 presence within haemocytes. Thus, clear distinction to the reader needs to be made that the presence is within haemocytes/immunocyte or is presently unknown. 
  17. Line 147-149 - this statement is the opinion of the author, but not datasets factual for inclusion in a review. This should be removed as it is the author's opinion that is a reflection of current paradigm, but is not factual/data driven. 
  18. Line 155-157 - citation or source is required for the statement by the authors
  19. Figure 2 - To ensure readers do not become confused, recommend changing the wording of Iba1-positive to Iba1/AIF1-positive to reinforce they are the same gene (and thus using same antibody to target the Iba1/AIF1 for immunostaining). 
  20. Line 237 - Should cite the paper that has investigated this (PMID 25569805)

GENERAL RECOMMENDATIONS

  1. The immunohistochemistry figure datasets are not valuable. There is no meaningful reason to include them. Furthermore, there are no materials/methods to the approach, reproducibility, etc. This is a review, not a primary research driven piece. It serves no purpose other than space filling. 
  2. A figure showing the metazoa phylogenetic tree of each of the groups in the manuscript (i.e. mollusca, annelida, cnidaria, porifera, etc.) in phylogenetic tree coupled with AIF1 percent homology. That is more valuable than some of the already published immunohistochemistry figures (i.e. for this reviewer, those immunohistochemistry figures provided nothing of value). This is important as the conclusion notes (Lines 219-221) The sequences 219 of the predicted proteins display a high level of conservation, particularly concerning the structure 220 of the two EF-hand Ca2+-binding regions

Minor points:

  1. Grammar and context to be updated (i.e. English fluency). Recommend having a native English speaker help smooth out the text
  2. (Line 10, 34, and other areas) - Just a little note -- in English -- it is not as fluid to put the word “The” in front of Allograft Inflammatory Factor-1, because it is already a personal noun. It is unnatural -- as saying “The David is a boy”.... As opposed to correctly “David is a boy”.
  3. (Line 62) - the word “respectively” is not necessary -- there are no two items initially compared to the trailing two items. I.e. there is nothing with respect to.
  4. Line 100 - not allogenic, the correct word is “allogeneic”. Much different meaning.

Reviewer 2 Report

Comments for Authors

The Manuscript ID: biology-933236 entitled “The allograft inflammatory factor-1 in Metazoans” present current information on the role and modulation of AIF-1, which is a highly conserved inflammation-responsive protein, and underlines its function along the evolutionary scale.

In vertebrates, AIF-1 (Allograft Inflammatory Factor 1), a calcium-binding protein, is involved in the activation of macrophages. Correspondingly, in invertebrates, the expression of the gene of this protein increases considerably after a bacterial challenge indicated that it plays a key role during the immune responses.

After a brief introduction, the authors report the knowledge on AIF-1 in Vertebrata and in phylogenetically distant group of species of Invertebrata (sponges, cnidarians, mollusks, annelids and echinoderms). Furthermore, the authors report the conclusions on the significance of AIF-1 in different cellular processes and highlighted the importance of to understand the molecular mechanisms in which AIF-1 is involved. This manuscript contributes to the development of an important area of research and it is important because it focuses mainly on invertebrates, important organisms in the context of immunobiology in relation to the simple mechanisms underlying their immune system. In particular, it takes into account the protein AIF 1 whose molecular characteristics and functions in invertebrates are the subject of rare investigation.

Comments (minor Revision)

Lane 2 - I found the title misleading. The work described the role and modulation of AIF-1 in Metazoa but it focuses more on Invertebrate. In my opinion, the title could be: “The allograft inflammatory factor-1 in Metazoans: focus on Invertebrates”.

Lane 25 Introduction - In my opinion is poorly treated: in this section, it is important to identify and justify the need for review.

Lanes 126-134 – Mollusca Since that the studies regarding characteristics and functions of AIF-1 in molluscs are limited I suggest to add this paper “Li et al., 2016 Characterization of allograft inflammatory factor-1 in Hyriopsis cumingii and its expression in response to immune challenge and pearl sac formation. Fish Shellfish Immunol. 59: 241-249”. This study reports the full-length cDNA of AIF-1 from H. cumingii (HcAIF-1) and investigates the temporal expression pattern of HcAIF-1 transcript under immune challenge and during pearl sac formation contributing to understand of the role of AIF-1 in innate immunity system of Invertebrates.

Lanes 150-167 – Annelida Regarding AIF-1 and Hirudo please cite recent papers of Baranzini et al. These papers suggested that AIF-1 and RNASET2 play a crucial role in orchestrating a functional cross-talk between granulocytes and macrophages in leeches, resulting in the activation of an effective response against pathogen infection.

Hirudo verbana as an alternative model to dissect the relationship between innate immunity and regeneration. Invertebrate  Survival Journal (2020). 17(1): 90-98.

AIF-1 and RNASET2 Play Complementary Roles in the Innate Immune Response of Medicinal Leech. J. Innate Immun. (2019). 11: 150-167.

Lanes 172-217 – Echinodermata Please cite a new paper of Chiaramonte et al., entitled “Allograft Inflammatory Factor AIF-1: early immune response in the Mediterranean sea urchin Paracentrotus lividus” in Zoology (2020) 142 – 125815. This paper reports the sequence of AIF-1 gene in Paracentrotus lividus, complete protein structure bioinformatically characterized and its expression, following Gram- LPS challenge at different stages.

Lanes 255-410 – References Please check correct Abbreviated Journal Names (for example Nature reviews. Neuroscience: will be Nat. Rev. Neurosci.)

Reviewer 3 Report

This article summarizes the fact that allograft inflammatory factor-1, a clacium-binding cytokine, is involved in inflammation in various metazoans phyla. There is nothing special or new in the content, but it is valuable as a review article in that it summarizes the role of each metazoans phyla very concisely. However, it seems that the depth is not so high as it does not focus on AIF-1 specific functions in the content or does not deal with the newly mentioned part of the content.

Round 2

Reviewer 1 Report

No additional comments. Authors have well addressed this reviewer's concerns and recommended changes. Subtle updates in grammar are recommended, but that can be left to the copy editors (i.e. nothing serious, just a few sections that do not sound so fluid in American English use).